# Satellite DNAs in Health and Disease

**DOI:** 10.3390/genes13071154

**Published:** 2022-06-26

**Authors:** Đurđica Ugarković, Antonio Sermek, Sven Ljubić, Isidoro Feliciello

**Affiliations:** 1Department of Molecular Biology, Ruđer Bošković Institute, Bijenička 54, HR-10000 Zagreb, Croatia; antonio.sermek@irb.hr (A.S.); sven.ljubic@irb.hr (S.L.); 2Department of Clinical Medicine and Surgery, School of Medicine, University of Naples Federico II, Via Pansini 5, 80131 Naples, Italy

**Keywords:** satellite DNA, heterochromatin, satellite RNA, epigenetics, heat stress, cancer, biomarker

## Abstract

Tandemly repeated satellite DNAs are major components of centromeres and pericentromeric heterochromatin which are crucial chromosomal elements responsible for accurate chromosome segregation. Satellite DNAs also contribute to genome evolution and the speciation process and are important for the maintenance of the entire genome inside the nucleus. In addition, there is increasing evidence for active and tightly regulated transcription of satellite DNAs and for the role of their transcripts in diverse processes. In this review, we focus on recent discoveries related to the regulation of satellite DNA expression and the role of their transcripts, either in heterochromatin establishment and centromere function or in gene expression regulation under various biological contexts. We discuss the role of satellite transcripts in the stress response and environmental adaptation as well as consequences of the dysregulation of satellite DNA expression in cancer and their potential use as cancer biomarkers.

## 1. Introduction

Satellite DNAs are tandemly repeated sequences preferentially located in the (peri)centromeric and subtelomeric heterochromatin of many eukaryotes, and they are clustered there in long megabase size arrays [1]. However, satellite repeats are also scattered throughout the euchromatin, often in the vicinity of genes or within introns, in the form of single repeats or short arrays [2]. Satellite repeats are usually located on all chromosomes and identical satellite DNA repeats on multiple chromosomes were shown to be important for the formation of the chromocenter and the maintenance of the entire genome in the nucleus [3,4]. The chromocenter is a cytological structure formed by sequence-specific DNA-binding proteins which cluster their cognate pericentromeric satellite DNA repeats and create physical links between heterologous chromosomes. Clustering of pericentromeric satellite repeats was shown to impose a physical barrier on homologous recombination, preventing the onset of chromosomal translocations [5] and the role of pericentromeric satellite DNA in higher order nuclear organization was also recently proposed [6]. Apart from the importance of pericentromeric satellite DNA in maintaining the nuclear structure and genome integrity, satellite DNAs are also characterized by high sequence divergence even among closely related species and by high homogeneity of repeats within each genome and species, suggesting their role in genome evolution and the speciation process [7,8,9,10,11]. In addition, recent studies revealed that the transcription of satellite repeats and their transcripts play a role in diverse cellular processes. In addition, the misregulation of satellite DNA transcription was shown to be associated with genomic instability and human diseases [2,12]. Considering the functional importance of satellite DNA transcription, this review preferentially deals with regulation of the expression of satellite DNA as well as the role of transcripts in processes related to health and disease.

## 2. Regulation of Satellite DNA Expression

Satellite DNA repeats located in (peri)centromeric heterochromatin are actively transcribed with transcripts being necessary for the formation and maintenance of heterochromatin as well as centromere function [12,13,14]. The presence of transcription factor binding sites was predicted and experimentally proven for different satellite DNAs [15,16] and transcription, which is often bidirectional, proceeds from internal promoters, usually by RNA polymerase II (RNA Pol II; [17,18,19,20,21]). Under physiological conditions, satellite DNA expression is low, and it is temporally and spatially regulated [18,22,23]. Small interfering RNAs (siRNAs) and PIWI-interacting RNAs (piRNAs), as final products of satellite DNA transcription in insects, nematodes and plants, are involved in the epigenetic regulation of transcription through a RNAi mechanism [24,25,26]. In *Drosophila melanogaster*, female germline satellite DNA-derived piRNAs are involved in heterochromatin establishment at their own genomic loci and satellite transcription is heterochromatin-dependent [27], while in *D. melanogaster* males, satellite-derived siRNA direct chromatin modification at 1.688 X chromosome satellite repeats, helping the dosage compensation machinery to identify the X chromosome [28]. In other insects, such as the beetle *Tribolium castaneum*, the major (peri)centromeric satellite DNA TCAST1 is expressed into piRNAs in the germline and into small interfering RNAs (siRNAs) in somatic cells [23]. TCAST1 piRNAs and TCAST1 siRNAs are involved in the establishment and maintenance of heterochromatin, respectively, acting *in cis* at the genomic loci from which they derive (Figure 1). It was proposed that the differential processing of TCAST1 transcripts is enabled by the existence of TCAST1 piRNA and siRNA-specified heterochromatic clusters whose expression is separately regulated; such “compartmentalization” might allow the same satellite DNA to respond specifically to different signals and to participate in multiple cellular processes [23]. 

In mammals, RNA Pol II transcribes the pericentromeric satellite DNA repeats into long non-coding RNA in a bidirectional fashion and dsRNAs can be recognized and possibly cleaved by Dicer1, which seems to control the level of satellite transcripts during mitosis [29,30]. In meiosis, however, the MIWI protein guided by piRNAs together with Dicer1 cleaves the access of satellite RNA and regulates its cellular level [31]. The mechanism by which Dicer regulates the transcription of pericentromeric satellite DNAs even in the absence of small RNAs, seems to be conserved from fission yeast to mammals [32]. Mouse pericentromeric major satellite DNA transcripts form RNA:DNA hybrids, which enable the retention of the Heterochromatin Protein 1 (HP1) proteins [33] and methyltransferases SUV39h1 and SUV39h2 [34,35], while enrichment of transcripts with m6A RNA modification is proposed to facilitate their association with heterochromatin [36]. The heterochromatic state of the pericentromere plays a role in recruiting and/or maintaining cohesin at the centromere to ensure the proper separation of sister chromatids [37]. In addition, specific microRNAs, such as miR-30a-3p, miR30d-3p and miR-30e-3p with complementarity to major mouse satellite DNA, probably guided by the Argonaute protein 1 (AGO1), are shown to play a role in the regulation of the expression of major satellite transcripts in mouse embryonic stem cells (mESC; [38]). Satellite DNA transcription in mammals is also important for early embryonic development as well as stem cell function and numerous transcription factors such as PAX3, PAX9 or FOXD3 play a role in the transcription regulation of the major mouse satellite DNA [39,40]. During early mouse embryogenesis, the functions of the two strands of major satellite RNA appear to be independent [41,42], with the forward strand directly involved in de novo targeting of the small ubiquitin-like modifier (SUMO)-modified HP1α to pericentromeric heterochromatin [33]. During the cell cycle, mouse pericentromeric major satellite transcripts accumulate primarily in the late G1 phase [17,42] and G1 arrest is a prerequisite for the entry of cells into the G0 phase [43]. Satellite DNA transcripts also seem to play an important role during insects’ embryogenesis. In the beetle *T. castaneum*, the burst of major TCAST1 satellite transcription occurs during early embryogenesis, coinciding with the initial establishment of constitutive heterochromatin [23], while in the mosquito *Aedes aegypti*, satellite-derived piRNAs participate in the degradation of transcripts during early embryogenesis [44].

The level of transcripts detectable within the active human centromere is low and is in contrast with the higher transcriptional levels of pericentromeric satellites [45], which are necessary for heterochromatin maintenance [34]. The predominant factor controlling human α satellite transcription seems to be the presence of centromere–nucleolar contacts and the transcripts are not exported to the cytoplasm, although they are not tightly bound to the centromere [21]. In the regulation of the transcription of centromeric α satellite DNA, the protein CENP-B is also involved [46]. Namely, CENP-B promotes the binding of the zinc-finger transcriptional regulator (ZFAT) which activates RNA Pol II transcription through histone modification H4K8Ac. α satellite RNA level fluctuates throughout the cell cycle, peaking in the G2/M phase [21], being controlled by the proteasome [16]. Satellite RNA interacts with members of the cohesin ring, suggesting a role for these transcripts in the regulation of mitotic progression [16]. The abundance of centromeric satellite RNAs seems to be regulated during the cell cycle in other organisms such as mouse, peaking in G2 phase, and long transcripts undergo post-transcriptional processing to generate smaller RNAs from 120 to 150 nucleotides [47]. Mouse centromeric transcripts are involved in the timely recruitment of the Chromosomal Passenger Complex (CPC) including AURORA B, INCENP and SURVIVIN just before the onset of mitosis [48]. Therefore, up or downregulation of centromeric satellite transcripts impairs cell cycle progression and has a detrimental effect on mitosis [49]. Recent results demonstrate that centromeric transcription, rather than centromeric transcripts itself, promotes centromeric cohesion in mitosis [50] and plays an important role in the deposition of the specific histone H3 variant CENP-A on centromeres during interphase [51]. The histone variant CENP-A is the epigenetic determinant for the centromere, where it is interspersed with canonical H3 to form a specialized higher order compact chromatin structure together with protein CENP-N [52].

Apart from CENP-A, centromeric and pericentromeric regions, both composed of satellite DNAs, are also characterized by different histone modifications: H3 lysine 4 and lysine 36 methylations (H3K4me1/2 and H3K36me1/2) at the centromere and H3K9me2/3 and H4K20me3 at the pericentromeric heterochromatin [53,54], which could affect the distinct regulation of satellite DNAs in the two domains. In addition, a hypoacetylated state at the centromeres, specifically H4K16ac and H3K4ac, seem to be conserved across eukaryotes [55,56], while at pericentromeric satellite repeats, SIRT6, a member of the Sirtuin deacetylases, maintains the silent state through the deacetylation of acetylated H3K18 (H3K18ac) [57]. In addition to the differences in histone modifications, the patterns of centromeric DNA methylation vary across different species and tissues [58], e.g., the centromeric satellite of mice contains 2-3 methylated CpGs *per* repeat unit, whereas the density of methylated CpGs is higher at the pericentromeres [12,59]. Recent detailed characterization of epigenetic patterns in human centromeres revealed a hypomethylated region, known as the centromeric dip region, embedded within a hypermethylated higher order repeat (HOR) of the α satellite that is occupied by CENP-A [60]. Also, α satellite arrays in the active centromere generally have higher CpG methylation compared with that of neighbouring inactive α arrays [61]. Plant centromeric satellites are differentially methylated: in rice and maize they are hypomethylated, while in *Arabidopsis* they are significantly methylated [62,63]. 

## 3. Satellite DNA in Stress Response 

Although it is evident that expression of satellite DNAs is tightly regulated under physiological conditions, under specific conditions and in several biological contexts their expression is significantly changed [2,12]. Heat stress (HS) specifically affects heterochromatin in different organisms: plants, insects, mice as well as in human cells by provoking its decondensation and decrease of nucleosome occupancy, resulting in transcription activation of heterochromatic satellite DNAs [20,64,65,66,67,68,69]. A very strong increase of pericentromeric satellite III (HSATIII) expression is induced by heat shock, DNA damaging agents and hyperosmotic stress [70], acting mostly through heat shock transcription factor 1 (HSF1), which binds satellite III DNA, recruits major cellular acetyltransferases to pericentromeric proteins and directs the recruitment of Bromodomain and Extra-Terminal (BET) proteins, BRD2, BRD3 and BRD4, which are required for satellite III transcription [71,72]. In addition to histone hyperacetylation in pericentromeric heterochromatin, the death domain-associated protein, DAXX, which acts as a chaperone for pericentromeric histone H3.3, also promotes HSATIII transcription by RNA Pol II of one strand [70]. While HSF1 affects satellite III transcription upon heat stress, tonicity enhancer binding protein (TonEBP) controls satellite III transcription in response to hyperosmotic stress [73]. It seems that stress-induced activation of satellite III is a part of the general cellular response to stress, which provides protection against heat-shock-induced cell death [74]. The molecular mechanism includes concentration of dephosphorylated splicing factors SRSF1 and SRSF9 as substrates for HSATIII RNAs and after stress removal, the enzyme CLK1 is recruited to rapidly rephosphorylate the pre-captured SRSFs [75]. In addition, nSBs sequester the m6A writer complex to methylate HSATIII, leading to subsequent sequestration of the nuclear m6A reader, YTHDC1. Sequestration of these factors from the nucleoplasm represses m6A modification of pre-mRNAs, leading to the repression of m6A-dependent splicing during the stress recovery phase [76]. Therefore, by different molecular mechanisms, satellite III RNA mediates the recruitment of a number of RNA-binding proteins involved in pre-mRNA processing and controls gene expression at the level of splicing regulation [75,76]. The alteration of splicing profiles is mainly characterized by an increase in intron retention events during the recovery from heat shock. Intron retention prevents the export of the pre-mRNAs from the nucleus, resulting in a suppression of gene expression at the posttranscriptional level. It was reported that satellite III and satellite II exhibit copy number variation (CNV) during the stress response, aging and pathology, and a close link between their transcription and CNV is postulated [77].

Activation of transcription of the major *T. castaneum* (peri)centromeric satellite DNAs, TCAST1 and human α satellite specifically, occurs upon heat stress [20,69]. In the case of TCAST1, it is coupled with satellite DNA demethylation [78], indicating the influence of DNA methylation on TCAST1 satellite expression. TCAST1 transcripts induced upon heat stress and their derived siRNAs as well as α satellite transcripts are proposed to play a role in heterochromatin maintenance and its recovery after HS by transiently increasing the level of silent histone modifications H3K9me2/3 at satellite repeats within heterochromatin [20,23,69]. In contrast to the major TCAST1 satellite, transcription of the minor *T. castaneum* satellite DNAs is not induced by HS, which can be related to their genome organization, characterized by preferential location in euchromatin [23,79]. Consequently, no significant change in the level of the silent histone mark H3K9me3 at minor satellite repeats is observed upon HS. In the case of the major TCAST1 satellite DNA, increased levels of H3K9me2/3 are detected after HS not only at regions of (peri)centromeric heterochromatin, but also at dispersed satellite repeats and their flanking regions up to 2 kb from the insertion site, indicating that TCAST1 siRNAs can act *in trans*, targeting homologous regions in euchromatin (Figure 1). Increased levels of H3K9me2/3 at euchromatic satellite repeats correlates with transient suppression of neighbouring genes and indicates a role for TCAST1 siRNAs in the modulation of gene expression [80]. 

## 4. Satellite DNA and Environmental Adaptation

There are a few explanations for the physiological consequence of satellite DNA activation followed by transient suppression of gene expression upon heat stress. Genomes undergo substantial transcriptional silencing upon heat stress and human satellite III participates in this process [74,75,76]. Differing from human satellite III RNA, which affects gene expression genome-wide, *T. castaneum* TCAST1 satellite influences the expression of genes located in the vicinity of euchromatic TCAST1 repeats [80]. Among these genes, a significant overrepresentation of immunoglobulin-like and developmental genes is found [81]. Mammalian genes involved in immunity and stress are more likely to contain transposons within UTRs [82], while plant genes for environmental response and development are enriched with introns composed of repeats which form a heterochromatic structure [83,84]. The association of particular tandem repeats involved in the silencing of the imprinted *SDC* gene in *Arabidopsis thaliana* with the expression of the same gene upon heat stress was reported [85]. These data indicate that transposons or satellite repeats are preferentially linked with environmentally susceptible genes, indicating possible influences on their expression. Human genome-wide analysis using ENCODE project data showed that not only (peri)centromeric satellite repeats, but also repeats located in euchromatin, are enriched in H3K9me3 and this histone modification is not limited to satellite DNA instances, but instead encompasses a wider region of flanking sequences [86]. The H3K9me3 level is also increased upon heat stress, as revealed for α satellite DNA repeats [69] (Figure 2). This association of euchromatic satellite repeats with repressive histone marks suggests their possible influence on the expression of neighbouring genes even under standard physiological conditions and, in particular, after heat stress. In addition, based on 3D genome structures, pericentromeric heterochromatin spatially contacts distant euchromatin regions enriched for repressive H3K9me2/3 marks in *D. melanogaster* [87], suggesting a possible influence on the expression of euchromatic genes. It was also shown that dispersed euchromatic satellite repeats can engage in homotypic interactions with identical repeats at pericentromeric heterochromatin, influencing the expression of genes proximal to euchromatic repeats, as revealed for the *bw* gene in *Drosophila* [88].

The transcription of satellite DNAs induced by different environmental stress conditions could be coupled to new satellite repeat insertions and changes in the copy number of repeats as well as their dispersion profiles [2], suggesting a mobile nature of satellite DNAs. Therefore, the movement of satellite DNA repeats provoked by environmental stress can promote genome change. This is evidenced by the existence of insertion polymorphism of euchromatic satellite DNA repeats among strains of the beetle *T. castaneum* [80,89]. The mobility of satellite DNA throughout the genome was proposed to occur by different mechanisms reviewed in [2], among them the rolling circle amplification, followed by site-specific recombination being the most probable one [90]. The increased activity of satellite DNAs in terms of transcription and spreading throughout the genome can provide genetic variability and gene expression divergence among populations and might play a role in the rapid response to stress and new environments. In particular, for species that have a high satellite DNA content, such as insects of the genus *Tribolium*, satellite repeats might influence adaptations to different habitats and environmental conditions. Variable occurrence of satellite DNA in different strains and isolates of the parasitic flatworm *Schistosoma mansoni* also points to the possible mobile nature of the satellite, affecting genetic variability and might be important for the evolution and biology of the species [91]. Satellite DNAs are known to be subjected to a high evolutionary turnover, resulting in not only a rapid copy number change [92,93,94], but also the emergence of new satellites [95]. One such example is the newly amplified satellite DNA in the New World Monkey genus *Aotus*, which enables night vision [96]. 

## 5. Satellite DNA in Oncogenic Transformation 

It is known that different pathological conditions can activate transcription of satellite DNAs. In diverse epithelial cancers such as the pancreas, lung, kidney, colon and prostate cancers, a significant increase in satellite DNA transcripts is detected [97]. However, the transcription profile of pericentromeric satellite DNAs is not only changed in solid tumors where it is attributed to cancer cells, but it is also changed in hematopoietic malignancies [98]. Namely, multiple myeloma transcriptomes are enriched in pericentromeric tandem repeat transcripts in cells of hematopoietic and non-hematopoietic origin, which include endothelial cells and mesenchymal stromal cells [99], and satellite transcription can be induced in healthy donors’ mesenchymal stromal cells by co-culturing with multiple myeloma cells [98]. Since the (peri)centromeric regions where satellite DNAs are preferentially located are epigenetically controlled, the lower level of repressive histone mark H3K9me3 detected at satellite repeats in cancer cell lines relative to the normal cells [86] (Figure 2) and global hypomethylation, which is characteristic of cancer cells, can be responsible for aberrant satellite DNA transcription [100]. It is also known that the lysine-specific demethylase 2A (KDM2A), which is specific for H3K36, is downregulated in prostate cancer and the KDM2A level is negatively correlated with pericentromeric heterochromatin transcription [101]. In addition, the misregulation of Polycomb repressive complexes, PRC1 and PRC2, which is common in many cancers, affects pericentromeric silencing [102]. Apart from epigenetic changes, overexpression of satellite DNA is often associated with a deficiency of tumor suppressor protein p53 which restrains the movement of repetitive elements [103]. Also, deficiency of the tumor suppressor BRCA1 impairs the integrity of constitutive heterochromatin and affects transcription of satellite DNA repeats [104]. Finally, overexpression and activation of heat shock transcription factor 1 (HSF1), which is observed in cancer cells, can be related to increased satellite DNA expression [72,105]. 

What is the role of increased levels of satellite transcripts in cancer? It was shown that overexpressed heterochromatic satellite RNAs bind BRCA1 and associated proteins that are important for the stability of the replication fork and induce DNA damage as well as genomic instability and promote breast cancer formation [106]. In addition, in mouse K-ras-mutated pancreatic precancerous tissues, transcripts of a major pericentromeric satellite DNA inhibit the DNA damage repair function of the YBX1 protein and accelerate tumor formation by acting as “intrinsic mutagens” [107,108]. Human satellite II transcripts expressed in cancer cells are immunogenic and activate the innate immune system to produce cytokines [109]. The same satellite II transcripts can cause repeat expansions at pericentric heterochromatin via aberrant RNA:DNA hybrid formation [110]. Human satellite II RNA also changes the distribution of CCCTC-binding factor (CTCF) on the genome and induces the senescence-associated secretory phenotype (SASP)-like inflammatory gene expression through the functional impairment of CTCF in senescent cells and provokes tumorigenesis through a pathway involving exosomes. This represents a novel mechanism of CTCF regulation by satellite II RNA during cellular senescence, which may contribute to the risk of tumorigenesis [111]. In addition, demethylated human satellite II and its transcripts sequester chromatin regulatory proteins, PRC1 and MeCP2, into abnormal nuclear bodies in cancer, compromising the epigenome [112]. In herpesvirus-infected cells, expression of human satellite II is also strongly induced by viral proteins, while viral protein expression and release of infectious particles is modulated by satellite II transcripts [113]. In zebrafish, hypomethylation of pericentromeric sequences and the subsequent derepression of satellite transcripts triggers an interferon response [114]. In general, overexpression of centromeric satellite DNAs promotes chromosome instability, which correlates with tumor metastasis [104,115]. Chromosome instability creates micronuclei and results in the presence of cytosolic DNA, which activates the cGas-STING pathway (cyclic GMP-AMP synthase stimulator of the interferon gene) [115]. This is detrimental for cancer cells that inactivate the STING pathway using different strategies [116].

It was noticed that heat stress (HS) conditions protect cells against the toxicity of chemotherapeutic drugs, most prominently the topoisomerase 2 (TOP2) inhibitor etoposide [117]. Recent results show that in response to heat stress, human satellite III RNA recruits topoisomerase IIa (TOP2A) to nuclear stress bodies and generates resistance against the TOP2A inhibitor etoposide in lung cancer [118]. Etoposide is frequently used to treat lung cancer and it temporarily stabilizes transiently induced DNA double-stranded breaks (DSB) created by TOP2A. The inability of TOP2A to form a complex with etoposide results in decreased DNA damage after treatment that impacts downstream DNA repair pathways. Etoposide resistance can be overcome by inhibiting human satellite III expression with epigenetic regulator BRD4 inhibitors [118]. The results show potential roles of satellite RNAs in cancer therapy resistance and suggest the therapeutic relevance of human satellite III RNA.

According to the current results, it is evident that satellite RNA can promote tumor progression by different mechanisms: inducing mutations [107], affecting epigenetic regulators [112], enhancing tumor cell proliferation [119], provoking inflammation [109,111] and cancer therapy resistance [118] or compromising genome integrity [106,120]. On the other hand, satellite transcripts can be recognized by the innate immune system, thus triggering an immune response [109,114]. In this way, they could prompt clearance of cancer cells by the immune system and curtail tumor growth (Figure 3).

## 6. Satellite DNAs and RNAs as Cancer Biomarkers

Since satellite DNA overexpression occurs in cancer tissues and their transcripts are released into the bloodstream, it is possible to use such circulating satellite RNAs as biomarkers for various types of cancers [97] (Table 1). However, the level of satellite RNA in the sera of cancer patients is low and the RNA is unstable; therefore, to reproducibly measure RNA levels, new sensitive methods which include droplet digital PCR (ddPCR) were developed. A satellite II RNA circulating in the blood serum quantified by the sensitive method of tandem repeat amplification by nuclease protection (TRAP) combined with droplet digital PCR (ddPCR) enabled discrimination of healthy controls from patients with pancreatic ductal carcinoma (PDAC) [121]. Increased levels of human satellite II circulating in the plasma of breast, gastric, lung and bile cancers as well as sarcoma and Hodgkin’s lymphoma was detected and could be used as a potential diagnostic marker [122]. It was also shown that patients with breast cancer and high relative levels of α satellite RNA in their breast tissues exhibit a 10- to 20-fold increased risk for the development of multiple cancers when harboring no BRCA-related clinical features [123].

Recently, it was shown for patients with metastatic prostate cancer that the increased level of satellite RNA is not only characteristic for cancer, but can be detected in peripheral tissues such as blood cells, serving as a diagnostic marker for metastatic castration-resistant prostate cancer as well as a marker for monitoring the progress of metastatic disease [124]. A mechanism underlying the increased blood α satellite RNA levels was proposed. Namely, exosomes which carry an excess of satellite RNA from prostate cancer could deliver RNA to blood cells and in addition, activate signalling pathways which can lead to the increased expression of satellite DNA in blood cells. In addition, it is also possible that circulating tumor cells (CTC), which are found in the blood of patients with metastatic prostate cancer [128], precipitate with blood cells and contribute to the increased level of α satellite RNA. In any case, this was the first demonstration of an aberrant level of satellite RNA level in a peripheral tissue of cancer patients. It seems therefore that not only serum or plasma-circulating satellite RNA, but also blood cellular satellite RNA could serve as an indicator of a specific stage of cancer as well as a method for monitoring the progress of disease.

Besides satellite RNAs, which are overexpressed in cancer and serve as cancer biomarkers, satellite DNA exhibits change in DNA methylation and copy number variation in different cancers [110] (Table 1). In ovarian carcinoma, hypomethylation of satellite II of chromosome 1 has been associated with tumor grade and identified as a marker of the risk of relapse [125]. Tumoral α satellite DNA hypomethylation level was found to be a prognostic parameter in patients with advanced gastric cancer [126]. Human satellite SST1 carries distinctive methylation and transcriptional profiles, including an enhancer embedded in each unit, and this is found only in specific arrays on chromosomes 19 and 4 [45]. These satellite arrays are hypervariable in the human population and alterations in their activity have been linked to cancer [127,129]. The detection of copy number variation within long satellite DNA arrays is relatively complex and often requires development of new assays [130], such as nanoplate-based digital PCR. Further studies are necessary to reveal satellite DNAs and RNAs as potential diagnostic, prognostic or therapeutic cancer biomarkers.

## 7. Future Directions

Identification of epigenetic marks for satellite DNAs located at the centromere, pericentromeric heterochromatin as well as at satellite repeats dispersed within the euchromatin is expected to explain chromatin “compartmentalization” and transcriptional regulation of these sequences. Also, the sequencing and assembly of long arrays of satellite DNAs using new technologies could give insight not only into their structure, organization, dynamics and evolution, but also the regulation of their expression. Further characterization of the transcriptional machinery affecting satellite DNAs and the processing of their transcripts into different forms of small and long RNAs as well as characterization of the proteins associated with them is expected to contribute to the understanding of the physiological role of satellite DNAs and their transcripts under standard conditions and in different diseases. Analysis of RNA modifications of satellite transcripts and their functional significance is an unexplored field of research. In addition, the study of spatial 3D organization of satellite DNAs could show if there is an interaction between heterochromatic and euchromatic satellite DNA repeats, which might influence their expression and genome-wide gene expression regulation. To better understand the role and regulation of satellite DNAs and their transcripts in different pathological processes, the development of new experimental tools for functional studies is needed. Further study of satellite DNAs and RNAs as diagnostic, prognostic or monitoring biomarkers is an area that may lead to novel parameters in diagnosis and treatment strategies.

## Figures and Tables

**Figure 1 genes-13-01154-f001:**
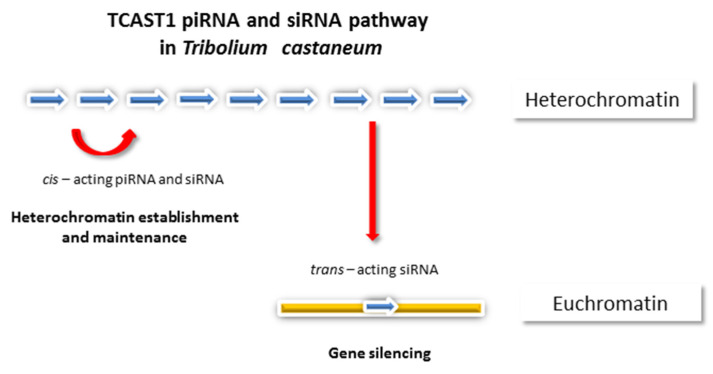
Role of major (peri)centromeric satellite DNA TCAST1 of beetle *Tribolium castaneum* in heterochromatin formation and maintenance as well as in gene silencing. TCAST1 transcripts are processed into piRNAs in germline and into siRNAs in the somatic cells. TCAST1 piRNAs and siRNAs are involved in the establishment and maintenance of heterochromatin, respectively, acting *in cis* at genomic loci from which they derive. The TCAST1 siRNA also acts *in trans*, affecting H3K9me3 level at euchromatic TCAST1 satellite elements and their neighbouring regions, influencing expression of genes located in the vicinity. The gene silencing effect is observed particularly upon heat stress coinciding with increased expression of TCAST1 satellite DNA. Blue arrows indicate TCAST1 satellite monomers.

**Figure 2 genes-13-01154-f002:**
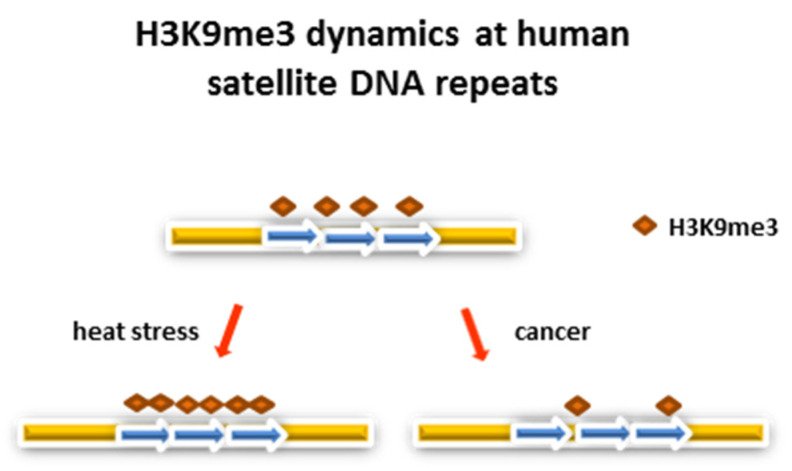
Dynamics of H3K9me3 level in human satellite DNA repeats. Enrichment of H3K9me3 at standard physiological conditions characterizes human satellite repeats located in both heterochromatin and euchromatin. Additionally, H3K9me3 level in satellite DNA repeats is enriched upon heat stress (HS), while in cancer cells the H3K9me3 level is decreased relative to normal cells. Blue arrows indicate satellite DNA monomers.

**Figure 3 genes-13-01154-f003:**
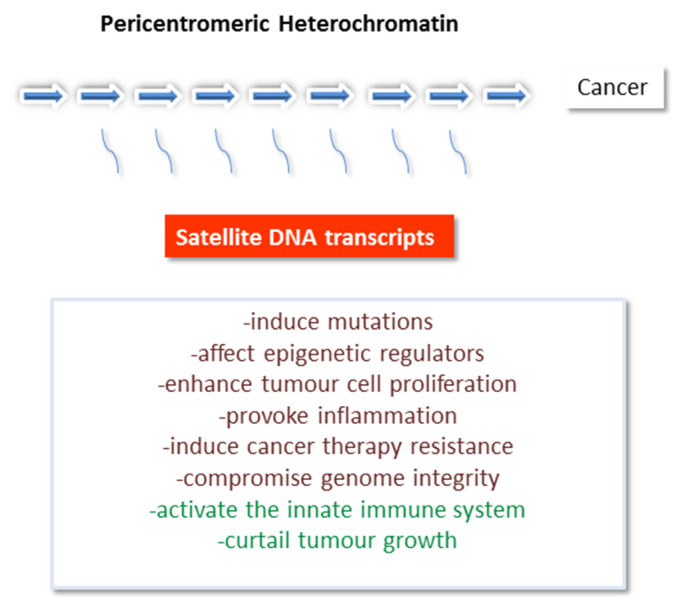
Transcription of pericentromeric satellite DNAs is significantly induced in different cancers and transcripts play diverse cellular roles which can promote cancer such as: inducing mutations, affecting epigenetic regulators, enhancing tumor cell proliferation, provoking inflammation, inducing cancer therapy resistance or compromising genome integrity. On the other hand, satellite transcripts can trigger the innate immune response and in this way, they could prompt clearance of cancer cells and curtail tumor growth.

**Table 1 genes-13-01154-t001:** List of satellite RNAs and DNAs associated with particular diseases for which they could serve as diagnostic or prognostic biomarkers.

Satellite DNA or RNA	Disease
Blood circulating satellite II RNA level	Pancreatic cancer [121], breast, gastric, lung cancers, sarcoma, Hodgkins’ lymphoma [122]
α satellite RNA level in cancer tissue	Breast cancer [123]
Blood cellular α satellite RNA level	Metastatic prostate cancer [124]
Hypomethylation of satellite II DNA	Ovarian cancer [125]
Hypomethylation of α satellite DNA	Gastric cancer [126]
Satellite SST1 activity	Colon cancer [127]

## Data Availability

Not applicable.

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
