# Peer review of "Satellite DNAs in Health and Disease"

_genes, 2022, doi:10.3390/genes13071154_

Round 1
Reviewer 1 Report
The manuscript by Ugarkovic and coworkers is an interesting review covering the functional roles of satellite DNA. It is well written and covers all aspects one may think about, with an adapted bibliography. I recommend its publication in the special issue devoted to non-coding DNAs and list below some remarks that may improve its quality.
p1 l30: not clear what authors mean by "complement"
p2 l71: not clear what a heterochromatic "cluster" is here.
p9 l373: the title of the 6th paragraph suggests that satellite DNAs can be used as biomarkers but the text focuses only on transcripts produced from satellite DNA.
p10 figure 4: this figure is not very informative. Moreover, it seems to describe an experimental protocol while the title points to a model that explains something, this is a bit disturbing. In the legend, shouldn't it be "cancer cells release exosomes" instead of "cancer releases exosomes"?
The paper contains many long sentences where addition of commas would facilitate understanding. Probably in some cases such commas are also grammatically required.
Author Response
The manuscript by Ugarkovic and coworkers is an interesting review covering the functional roles of satellite DNA. It is well written and covers all aspects one may think about, with an adapted bibliography. I recommend its publication in the special issue devoted to non-coding DNAs and list below some remarks that may improve its quality.
p1 l30: not clear what authors mean by "complement"
“Complement” is deleted.
p2 l71: not clear what a heterochromatic "cluster" is here.
Small RNAs, piRNAs and siRNAs derive from clusters which are in the case of TCAST1 satellite located in heterochromatin and we call them heterochromatic clusters.
p9 l373: the title of the 6th paragraph suggests that satellite DNAs can be used as biomarkers but the text focuses only on transcripts produced from satellite DNA.
We changed the title into Satellite DNAs and RNAs as cancer biomarkers. The paragraph deals with transcripts but also with methylation status of satellite DNAs and their activity which also serve as biomarkers.
p10 figure 4: this figure is not very informative. Moreover, it seems to describe an experimental protocol while the title points to a model that explains something, this is a bit disturbing. In the legend, shouldn't it be "cancer cells release exosomes" instead of "cancer releases exosomes"?
We deleted Figure 4. Instead we added Table 1 with list of representative satellite RNAs and DNAs which can serve as potential disease diagnostic or prognostic biomarkers.
The paper contains many long sentences where addition of commas would facilitate understanding. Probably in some cases such commas are also grammatically required.
Reviewer 2 Report
Comments on genes-1782163
In this review, the authors focused on recent progress related to the regulation of satellite DNA expression and the role of their transcripts under various contexts. This is indeed an interesting topic with few review papers published in recent years. This manuscript was written with a well-organized structure. Recent publications in 2021 and 2022 were also included and discussed. Overall, this is a review paper of good quality. The authors mentioned in the “Future directions” section that some developments/improvements are needed for this research area, but it is relatively difficult to find relevant information in the above sections. It is recommended that some revisions should be made.
1. It would be more informative if the authors could provide a section (or at least a table) listing the common techniques used to characterize satellite DNAs.
2. It would be more informative if the authors could provide a section (or at least a table) listing some representative applications of satellite DNAs for diagnosis/prognosis/monitoring biomarkers with key references.
Author Response
In this review, the authors focused on recent progress related to the regulation of satellite DNA expression and the role of their transcripts under various contexts. This is indeed an interesting topic with few review papers published in recent years. This manuscript was written with a well-organized structure. Recent publications in 2021 and 2022 were also included and discussed. Overall, this is a review paper of good quality. The authors mentioned in the “Future directions” section that some developments/improvements are needed for this research area, but it is relatively difficult to find relevant information in the above sections. It is recommended that some revisions should be made.
- It would be more informative if the authors could provide a section (or at least a table) listing the common techniques used to characterize satellite DNAs.
The review is focused on the regulation of satellite DNA transcription and functional significance of transcripts and we do not think that it is necessary to list techniques used to characterize satellite DNA. In the „Future directions“ we mentioned „sequencing and assembly of long arrays of satellite DNAs using new technologies“ as an important goal in satellite DNA research because it could give insight not only into their structure and organization but also in the regulation of their expression and potential function of transcripts. Besides, there are recent reviews (e.g. Šatović Vukšić and Plohl 2021, Prog. Mol. Subcell. Biol.) dealing with techniques and methods for satellite DNA characterization in pre-genomic and genomic era.
- It would be more informative if the authors could provide a section (or at least a table) listing some representative applications of satellite DNAs for diagnosis/prognosis/monitoring biomarkers with key references.
We added Table 1 with list of representative satellite RNAs and DNAs which can serve as potential diagnostic and prognostic biomarkers (with references).